# Methionine Supplementation Benefits Lipogenesis in Goat Intramuscular Adipocytes, Likely by Inhibiting the Expression of *SLC5A5*

**DOI:** 10.3390/ani15233450

**Published:** 2025-11-29

**Authors:** Jin Pan, Chengsi Gong, Xuening Li, Yanyan Li, Jiani Xing, Yaqiu Lin, Youli Wang

**Affiliations:** 1Key Laboratory of Qinghai-Tibetan Plateau Animal Genetic Resource Reservation and Utilization, Ministry of Education, Chengdu 610041, China; 19822962583@163.com (J.P.); 80300209@swun.edu.cn (C.G.);; 2Key Laboratory of Sichuan Province for Qinghai-Tibetan Plateau Animal Genetic Resource Reservation and Exploitation, Chengdu 610041, China; 3College of Animal & Veterinary Science, Southwest Minzu University, Chengdu 610041, China

**Keywords:** intramuscular adipocyte, methionine, adipogenesis, goat, *SLC5A5*

## Abstract

Intramuscular fat is a key determinant of meat quality. Numerous studies indicate that dietary methionine (Met) supplementation increases body fat in both humans and livestock. In this study, goat intramuscular adipocytes were cultured in basal complete medium supplemented with Met at 0, 50, 100, 200, 400, and 800 μM. This study demonstrates a quadratic relationship between the level of Met supplementation and cellular proliferation and differentiation, with an optimal effect observed at 100 μM Met. RNA-seq analysis showed that Met supplementation led to the downregulation of pathways related to lipolysis, such as the regulation of lipolysis in adipocytes, thyroid hormone synthesis, circadian rhythm, and circadian entrainment. Furthermore, the addition of Met led to the downregulation of gene expression in pathways involved in lipid metabolism, including the *PER2*, *PRKG1*, and *SLC5A5* genes. Through overexpression and inhibition of *SLC5A5*, it was discovered that *SLC5A5* suppresses the differentiation of intramuscular adipocytes. Subsequent research further indicated that Met supplementation promotes adipogenesis by inhibiting *SLC5A5*.

## 1. Introduction

Intramuscular fat (IMF) is the adipose tissue deposited in the muscle and plays an important role in assessment of meat quality. Typically, high content of IMF improves gustatory attributes, including tenderness and juiciness [1]. Recent surveys have shown that high IMF content in mutton or beef is more popular with consumers [2,3]. So, how to improve IMF is currently a focus of concern.

Lipogenesis is associated with the proliferation and differentiation of preadipocytes [4,5]. The differentiation of intramuscular adipocytes is similar to that of other fat tissues, such as abdominal fat and subcutaneous fat [6]. *C/EBPβ* is a crucial early factor in adipogenesis. During the initial stages of fat formation, *C/EBPβ* levels progressively increase, subsequently binding to the promoters of *PPARγ* and *C/EBPα* to activate their expression [7]. *PPARγ* and *C/EBPα* are key regulators that promote the differentiation of mesenchymal precursor cells into mature adipocytes, while also controlling triglyceride (TG) accumulation to increase cell volume [8,9]. Moreover, *GLUT4* and *FABP4*, which are expressed in adipocytes, encode proteins participating in fat deposition through their roles in fatty acid uptake, transport, and metabolism [10,11,12,13]. Fat is stored in adipocytes in the form of TG. The TG content within adipocytes is in a dynamic equilibrium and is regulated by both lipogenesis and lipolysis. It is well established that *ATGL* is the key enzyme involved in lipolysis [14]. So, the balance between lipogenesis and lipolysis in adipocytes determines the fat deposition.

Recently, an increasing number of studies have been conducted to explore enhancements in IMF. For example, researchers have shown that dietary supplementation of copper sulfate pentahydrate, resveratrol, vitamin A, and rumen-protected fat can increase the IMF content in cattle and goats [3,15,16,17]. Moreover, increasing studies have demonstrated that methionine (Met) is involved in fat accumulation. For instance, it has been shown that low dietary Met results in low body weight and fat mass in mice [18], which may be explained by upregulated fatty acid oxidation, glycolysis, and tricarboxylic acid cycle metabolism [19]. In addition, it has been documented that Met deficiency inhibits TG synthesis and increases lipogenesis in the mouse liver [20]. Furthermore, supplementation with Met in the diet of meat ducks significantly reduced abdominal fat, with a negative correlation observed between Met supplementation levels and abdominal fat. This mechanism operates by promoting the expression of hormone-sensitive lipase (*HSL*), thereby enhancing lipolysis [21]. *HSL* is a pivotal enzyme mediating triglyceride hydrolysis in hormone-regulated lipolysis [22]. Concurrently, studies indicate that Met supplementation regulates subcutaneous adipocytes in dairy cattle via oxidative pathways, yet merely maintains normal metabolic function without enhancing lipid accumulation in subcutaneous fat [23]. However, the impact of Met supplementation on cell proliferation and differentiation of the intramuscular preadipocytes of goats remains uncertain.

So, this study aimed to investigate the effects of different concentrations of Met supplementation on cell proliferation, cell differentiation, and cell viability in goat intramuscular preadipocytes. Then, given that the mechanism of Met on adipogenesis is currently still controversial [24], RNA-Seq was used to reveal the underlying mechanism.

## 2. Materials and Methods

### 2.1. Isolation of Goat Intramuscular Preadipocytes

All procedures involving animals were approved by the Institutional Animal Care and Use Committee of Southwest Minzu University (Protocol No. SMU-202401047). Longissimus thoracis et lumborum muscle tissue was collected from a newborn Jianzhou Big Ear goat immediately after slaughter. The tissue was minced and washed three times with phosphate-buffered saline (PBS) and then digested with collagenase type II (Solarbio, Beijing, China) at 37 °C for 90 min. Digestion was terminated by adding basal complete medium consisting of DMEM/F-12 (Gibco, Carlsbad, CA, USA) supplemented with 10% fetal bovine serum (Gibco, CA, USA). The digestion mixture was filtered sequentially through 200- and 400-mesh sieves, and the filtrate was centrifuged at 2000 rpm for 5 min. The cell pellet was resuspended in basal complete medium and seeded in 100 mm culture dishes at 37 °C in a humidified 5% CO_2_ incubator. After 2 h of culture, adherent cells were collected and cryopreserved in liquid nitrogen for later use [25]. The final Met concentration in the basal complete medium was approximately 116.2 μM.

### 2.2. Cell Culture and Differentiation

Frozen intramuscular preadipocytes were thawed and seeded in 24-well plates (1 mL cell suspension containing 2 × 10^4^ cells per well). When cells reached approximately 90% confluence, differentiation was induced. Cells were cultured in basal complete medium supplemented with Met at 0, 50, 100, 200, 400, or 800 μM, with three biological replicates per treatment. Differentiation was induced by adding 0.5 mM isobutylmethylxanthine, 1 μM dexamethasone, and 10 μg/mL insulin for 2 d, followed by maintenance in 10 μg/mL insulin for an additional 2 d. Cells were then cultured in basal complete medium for a further 2 d.

### 2.3. Triglyceride Assay

Following differentiation, triglyceride (TG) content was quantified using a commercial assay kit (Applygen, Beijing, China). Cells were lysed and centrifuged at 12,000 rpm for 1 min, and the supernatant was heated at 70 °C for 10 min, followed by centrifugation at 2000 rpm for 5 min. TG concentration in the supernatant was measured spectrophotometrically at 550 nm.

### 2.4. Oil Red O Staining

Cells were washed three times with PBS and fixed in 4% paraformaldehyde for 30 min. Fixed cells were stained with Oil Red O solution (Solarbio, Beijing, China) for 1 h and then washed with PBS. Images were captured using an epifluorescence microscope (Olympus IX-73, Tokyo, Japan). For quantification, intracellular Oil Red O dye was extracted with 200 μL of isopropanol, and absorbance was measured at 490 nm.

### 2.5. Bodipy Staining

Cells were washed 3 times with PBS and then fixed in 4% formaldehyde for 30 min. The formaldehyde was discarded, and then the cells were washed with PBS 3 times. Then, 1 μL of the stored Bodipy stock solution was diluted in 1 mL of PBS to prepare the Bodipy working solution. Then, the cells were stained with the Bodipy working solution for 1 h (Solarbio, Beijing, China) and washed 3 times with PBS. Stained cells were photographed with an epifluorescence-equipped microscope (Olympus IX-73, Tokyo, Japan).

### 2.6. Crystal Violet Staining

Cells were seeded in 96-well plates (200 μL cell suspension containing 2 × 10^3^ cells per well) and cultured with basal complete medium supplemented with various Met concentrations. Each treatment had five replicate wells. After 0, 24, 48, and 72 h, cells were fixed with 4% paraformaldehyde for 12 h and stained with crystal violet solution (0.05 g crystal violet in 10 mL of ethanol, diluted 1:5 in PBS). After 20 min of staining, excess dye was removed, and cells were washed three times with PBS, air-dried, and imaged under a microscope (Olympus IX-73, Tokyo, Japan).

### 2.7. EdU Proliferation Assay

Goat intramuscular preadipocytes were seeded in 24-well plates (2 × 10^4^ cells per well) and cultured for 48 h. EdU solution (BeyoClick™ EdU Cell Proliferation Kit, Beyotime, Shanghai, China) was added to a final concentration of 20 μM and incubated at 37 °C for 2 h. Cells were fixed with 4% paraformaldehyde for 15 min, permeabilized with 0.3% Triton X-100 for 15 min, and stained according to the manufacturer’s protocol. Images were obtained using an epifluorescence microscope (Olympus IX-73, Tokyo, Japan).

### 2.8. Plasmid Construction, siRNA Synthesis, and Transfection

The coding sequence (CDS) of goat *SLC5A5* was amplified from cDNA and cloned into the pcDNA3.1(+) vector (Thermo Fisher Scientific, Waltham, MA, USA) using NheI and XhoI restriction sites to generate the overexpression construct (SLC5A5-OVER). The recombinant plasmid was confirmed by restriction enzyme digestion and DNA sequencing. The empty pcDNA3.1(+) plasmid served as a negative control.

Small interfering RNAs (siRNAs) targeting goat *SLC5A5* mRNA and a non-targeting control (siRNA-NC) were synthesized by GenePharma (Shanghai, China). The sequences were as follows:siRNA-NC:sense 5′-UUCUCCGAACGUGUCACGUTT-3′;antisense 5′-ACGUGACACGUUCGGAGAATT-3′.siRNA-SLC5A5-345:sense 5′-GCACCUACGAGUACCUGGA-3′;antisense 5′-UCCAGGUACUCGUAGGUGC-3′.

For transfection, intramuscular preadipocytes were seeded to reach 70–80% confluence and transfected with either plasmid DNA or siRNA using TurboFect Transfection Reagent (Thermo Fisher Scientific, MA, USA) according to the manufacturer’s protocol. After 16 h, the medium was replaced with either basal complete medium or medium supplemented with 100 μM Met.

### 2.9. RNA Extraction and Quantitative Real-Time PCR (qRT-PCR)

Total RNA was extracted using RNAiso Plus reagent (Takara, Kyoto, Japan). RNA purity and concentration were determined using a NanoDrop 2000 spectrophotometer (Thermo Fisher Scientific, MA, USA). Complementary DNA (cDNA) was synthesized using the PrimeScript RT Reagent Kit (TAKARA, Kyoto, Japan) and stored at −30 °C. qRT-PCR was conducted as described previously [26], using gene-specific primers designed based on sequences from the NCBI database (synthesized by Sangon Biotech, Shanghai, China). *GAPDH* was used as the reference gene, and relative expression levels were calculated using the 2^−(∆∆Ct)^ method (Table 1).

### 2.10. RNA Sequencing and Bioinformatics Analysis

Intramuscular adipocytes were cultured for 72 h in basal complete medium (control, *n* = 3) or medium supplemented with 100 μM Met (*n* = 3). Total RNA was extracted using TRIzol reagent (Invitrogen, Carlsbad, CA, USA) and treated with RNase-free DNase I (TAKARA, Kyoto, Japan). RNA integrity was confirmed using an Agilent 2100 Bioanalyzer (Agilent Technologies, Santa Clara, CA, USA), and concentrations were determined by Nanodrop (TY20190063.IMPLEN, Munich, Germany).

Sequencing libraries were prepared using the NEBNext^®^ Ultra™ RNA Library Prep Kit for Illumina^®^ (Nebraska, NE, USA) following the manufacturer’s instructions. Libraries were sequenced on an Illumina NovaSeq 6000 platform (Allwegene Technology Co., Beijing, China) to generate 150 bp paired-end reads. Differentially expressed genes (DEGs) (Appendix A) were identified using DESeq2 with an adjusted *p* < 0.05 and |log_2_(−foldchange)| ≥ 0.7. Gene Ontology (GO) enrichment (Appendix A) analysis was performed using the GOseq R 4.5.2 package [27], and Kyoto Encyclopedia of Genes and Genomes (KEGG) pathway enrichment (Appendix A) was analyzed using KOBAS 3.0 software [28].

### 2.11. Statistical Analyses

All experiments were carried out in three biological replicates and repeated three times. Data were analyzed by SPSS (Version 25.0, SPSS Inc., Chicago, IL, USA). Statistical differences with 3 or more treatments were determined by one-way ANOVA, and differences among treatments were determined by Duncan’s multiple range test. In addition, orthogonal polynomials were used to evaluate linear and quadratic responses. The difference between 2 treatments was determined by Student’s *t*-test. Labeled means without a common letter are significantly different; * means *p* < 0.05, ** means *p* < 0.01, and different letters on the bar graph indicate significant differences, *p* < 0.05 [29].

## 3. Results

### 3.1. Impact of Supplementation with Met on Cell Proliferation of Intramuscular Preadipocytes

Triglyceride deposition is determined by both the number and differentiation status of adipocytes. Therefore, we first investigated the proliferative capacity of intramuscular preadipocytes in response to Met supplementation. The results demonstrated that the cell number increased quadratically with rising levels of Met supplementation (*p* = 0.012; Figure 1A,B), with the greatest cell number observed at 100 μM Met. To further assess cell proliferation, 5-ethynyl-2′-deoxyuridine (EdU), a thymidine analog that incorporates into cellular DNA and reacts with a fluorescent azide through a copper-catalyzed cycloaddition, was employed [30]. The number of EdU-positive cells increased linearly with elevated Met supplementation (*p* < 0.001; Figure 1C), and the proportion of EdU-positive cells followed a similar linear trend (*p* < 0.001; Figure 1D). These findings indicate that Met supplementation at 100 μM effectively promotes the proliferation of intramuscular preadipocytes.

### 3.2. Impact of Supplementation with Met on Cell Differentiation and Lipid Metabolism of Intramuscular Preadipocytes

To investigate the effect of Met on the differentiation of intramuscular preadipocytes, cells were treated with various concentrations of Met and subsequently stained with Oil Red O. The results revealed significant linear (*p* < 0.001) and quadratic (*p* < 0.001) effects with increasing Met supplementation (Figure 2A,B). Similarly, intracellular triglyceride (TG) content exhibited both linear (*p* < 0.001) and quadratic (*p* = 0.001) responses, with the greatest TG accumulation observed at 100 μM Met (Figure 2C). When Met supplementation exceeded 200 μM, TG deposition gradually declined. These results indicate that 100 μM Met supplementation most effectively promotes the differentiation of intramuscular preadipocytes.

Combining these findings with the proliferation data, we confirmed that Met supplementation at 100 μM enhances intramuscular fat deposition. To further elucidate the molecular mechanisms, we analyzed the expression of genes associated with adipogenic differentiation. Met supplementation significantly upregulated *PPARγ* expression (*p* = 0.003; Figure 2D) and *FABP4* expression (*p* = 0.012; Figure 2E), the latter being a well-known marker of triglyceride accumulation [31]. Conversely, *ATGL* expression, a key gene involved in intracellular triglyceride hydrolysis [32,33], was significantly downregulated by Met supplementation (*p* = 0.029; Figure 2F).

### 3.3. Impact of Supplementation with Met on Survival Rate of Intramuscular Preadipocytes

Cell viability under basal and Met supplementation conditions (100 μM) was assessed using crystal violet staining. The results indicated that differences in cell viability emerged after 48 h of incubation (*p* < 0.05) and became more pronounced with extended incubation time (*p* < 0.01; Figure 1E,F). Overall, Met supplementation at 100 μM enhanced cell viability compared with cells cultured in the basal complete medium, particularly after 72 h of incubation.

To further evaluate the potential cytotoxic effects of Met, cell viability was also measured using the CCK-8 assay. The results demonstrated a linear decrease in cell survival rate with increasing concentrations of Met, with the highest survival observed at 100 μM supplementation (*p* < 0.001; Figure 1G). Notably, even at concentrations up to 800 μM, Met supplementation did not significantly reduce cell viability compared with the basal medium (Figure 1G), suggesting that elevated Met levels exert no cytotoxic effects on intramuscular preadipocytes.

### 3.4. RNA-Seq Analysis

To investigate the molecular mechanisms underlying the effects of methionine (Met) supplementation on intramuscular preadipocyte development, RNA sequencing (RNA-seq) was performed on cells cultured in basal medium (control, CT) and medium supplemented with 100 μM Met. Principal component analysis (PCA) revealed a clear separation of transcriptional profiles between the CT and Met-treated samples, indicating substantial transcriptomic differences (Figure 3A). Compared with the CT group, a total of 95 differentially expressed genes (DEGs; adjusted *p* ≤ 0.05, |log_2_(−fold change)| ≥ 0.7) were identified in response to Met supplementation (Figure 3B). Among these, 44 genes were upregulated and 51 genes were downregulated in the Met-treated group relative to the control (Figure 3B,C).

### 3.5. Enrichment Analysis of the Differentially Expressed Genes

Gene Ontology (GO) enrichment analysis assigned the differentially expressed genes (DEGs) to 2071 terms encompassing biological processes and molecular functions. The top 30 significantly enriched GO terms for upregulated and downregulated genes are presented in Figure 4A,B. Upregulated DEGs were predominantly associated with developmental growth, regulation of developmental growth, sarcomere organization, regulation of heart growth, myofibril assembly, DNA-binding domain interactions, ligase regulator activity, and activin receptor antagonist activity, among others (Figure 4A). Conversely, downregulated DEGs were enriched in terms related to regulation of anatomical structure size, positive regulation of natural killer cell-mediated cytotoxicity, positive regulation of defense response, positive regulation of leukocyte-mediated cytotoxicity, regulation of natural killer cell-mediated immunity, phytoceramidase activity, and sodium:iodide symporter activity (Figure 4B).

Kyoto Encyclopedia of Genes and Genomes (KEGG) pathway analysis revealed that DEGs were assigned to 63 pathways, with the top 20 enriched pathways shown in Figure 4C,D. Upregulated pathways included mannose-type O-glycan biosynthesis, graft-versus-host disease, N-glycan biosynthesis, allograft rejection, type I diabetes mellitus, autoimmune thyroid disease, acute myeloid leukemia, renal cell carcinoma, antigen processing and presentation, ErbB signaling, axon guidance, T cell receptor signaling, and human immunodeficiency virus 1 infection, among others (Figure 4C). Downregulated pathways involved circadian rhythm, vasopressin-regulated water reabsorption, sphingolipid metabolism, circadian entrainment, regulation of lipolysis in adipocytes, long-term depression, adherens junction, thyroid hormone synthesis, and steroid hormone biosynthesis (Figure 4D).

### 3.6. Met Supplementation Influence on the Pathways Associated with Adipogenesis

Among the altered KEGG pathways, we focused on those associated with adipogenesis, including ErbB signaling, thyroid hormone synthesis, regulation of lipolysis in adipocytes, circadian rhythm, and circadian entrainment. In the ErbB signaling pathway, *PAK6* was identified as a key gene and was upregulated in this study (Figure 4E). Previous studies have reported that inhibition of adipogenic differentiation is accompanied by downregulation of ErbB signaling [34]. Within the thyroid hormone synthesis pathway, *SLC5A5,* which functions in brown and/or white adipose tissue thermogenesis [35], was significantly downregulated in Met-supplemented cells (Figure 4E), suggesting decreased lipolytic activity. Similarly, *PRKG1*, the key gene in the regulation of lipolysis in the adipocyte pathway, was significantly downregulated (Figure 4E); this gene has been reported to inhibit rat adipocyte proliferation via the cGMP-PKG signaling pathway [36]. In circadian rhythm and circadian entrainment pathways, *PER2*, which negatively regulates cell proliferation and adipogenic differentiation [37,38], was also downregulated (Figure 4E). These results suggest that Met supplementation modulates adipogenesis by influencing multiple signaling pathways, including those that regulate lipid metabolism and circadian biology.

### 3.7. Effect of Met Supplementation Following SLC5A5 Silencing on Goat Intramuscular Adipocyte Differentiation

Among the DEGs associated with adipogenesis, we focused on *SLC5A5*. Quantitative real-time PCR (qRT-PCR) confirmed that 100 μM Met supplementation significantly downregulated *SLC5A5* expression in goat intramuscular adipocytes (*p* < 0.05; Figure 5A), consistent with RNA-seq results. Silencing of *SLC5A5* significantly suppressed its expression (*p* < 0.01; Figure 5B). Functional assays revealed that *SLC5A5* inhibition increased both the size and number of lipid droplets, as evidenced by Oil Red O and BODIPY staining (*p* < 0.05; Figure 5C), which was corroborated by optical density measurements (*p* < 0.05; Figure 5D). Furthermore, *SLC5A5* silencing significantly upregulated the mRNA expression of key adipogenic markers, including *PPARγ*, *C/EBPα*, and *C/EBPβ* (*p* < 0.05; Figure 5E–H). Notably, Met supplementation following *SLC5A5* interference further enhanced lipid droplet accumulation (*p* < 0.05; Figure 5C) and further elevated the expression of *PPARγ*, *C/EBPα*, and *C/EBPβ* (*p* < 0.05; Figure 5E–H), indicating of additive effect of Met in promoting adipogenic differentiation through *SLC5A5* inhibition.

### 3.8. The Effect of Met Supplementation on the Differentiation of Goat Intramuscular Adipocytes Following SLC5A5 Overexpression

QRT-PCR confirmed a significant increase in *SLC5A5* mRNA expression, validating the successful construction of the *SLC5A5* overexpression plasmid (*p* < 0.05; Figure 6A). Functional assays demonstrated that *SLC5A5* overexpression significantly reduced lipid droplet accumulation in intramuscular adipocytes, as observed by Oil Red O and BODIPY staining (*p* < 0.05; Figure 6B) and corroborated by optical density measurements (*p* < 0.05; Figure 6C). Additionally, the mRNA expression levels of *GLUT4*, *C/EBPβ*, and *PPARγ* were significantly decreased following *SLC5A5* overexpression (*p* < 0.05; Figure 6D–G). Notably, methionine supplementation following *SLC5A5* overexpression mitigated these inhibitory effects, restoring lipid droplet accumulation (*p* < 0.05; Figure 6B) and significantly increasing the expression of *GLUT4*, *C/EBPβ*, and *PPARγ* (*p* < 0.05; Figure 6D–G). These findings indicate that Met supplementation can counteract the suppressive effect of *SLC5A5* overexpression on adipogenic differentiation in goat intramuscular adipocytes.

## 4. Discussion

Intramuscular fat (IMF) is a critical determinant of meat quality, particularly influencing taste and flavor. The proliferation and differentiation of intramuscular preadipocytes directly regulate fat accumulation and, consequently, IMF content. Previous studies have demonstrated that low methionine (Met) intake results in reduced fat mass in mice and humans [24]. This study explored the potential mechanisms by which Met supplementation promotes proliferation and differentiation of intramuscular adipocytes in goats.

IMF content is primarily determined by the number and size of intramuscular adipocytes, with apoptosis and proliferation playing key regulatory roles [39]. Met deficiency has been shown to suppress cell proliferation in porcine preadipocytes [40] and glioblastoma cells [41]. Similarly, studies in bovine mammary epithelial cells indicate that Met supplementation at 600 μM enhances proliferation, whereas concentrations above 600 μM gradually reduce this effect [21]. Consistently, our results revealed a quadratic relationship between adipocyte proliferation and Met concentration, with optimal effects observed at 100 μM. Notably, Met supplementation decreased *PRKG1* expression, a gene reported to inhibit adipocyte proliferation [36], which may partly explain the observed increase in proliferation. Furthermore, *PER2*, a key circadian rhythm gene negatively associated with cell proliferation [37,38], was downregulated following Met supplementation, reinforcing the proliferative effect of Met on goat intramuscular adipocytes.

Adipocyte differentiation is governed by both hyperplasia and hypertrophy [42,43] and is orchestrated by transcription factors including *C/EBPβ*, *C/EBPα*, and *PPARγ* [44,45]. Lipid metabolism is further regulated by lipolytic genes, such as *ATGL* and *LPL*, and lipid deposition genes, including *FABP4* and *GLUT4* [32,33,46]. In this study, Met supplementation exerted a quadratic effect on adipogenic differentiation, with 100 μM yielding maximal triglyceride accumulation and lipid droplet formation, as evidenced by Oil Red O staining and TG quantification. This effect was supported by upregulation of *FABP4* and *PPARγ* and downregulation of *ATGL*.

Transcriptomic analysis revealed that Met supplementation modulates key signaling pathways involved in adipogenesis, including ErbB signaling, thyroid hormone synthesis, lipolysis regulation, and circadian rhythm. The upregulation of *PAK6* in the ErbB pathway aligns with prior studies showing that ErbB signaling promotes adipogenic differentiation [26]. *SLC5A5*, a key gene in the thyroid hormone synthesis pathway, acts on the brown and/or white adipose tissues to generate heat [47]. Adipose tissue, as a major immune organ, activates its resident immune cells, such as macrophages and T cells, which release pro-inflammatory factors like TNF-α and IL-6, thereby inhibiting adipogenesis [48,49,50]. Supplementing with Met activates transplant-rejection-associated pathways, effectively enhancing T-cell clearance of “abnormal immune cells”. This reduces pro-inflammatory macrophage infiltration in IMF and alleviates inflammation’s inhibition of adipogenic differentiation. Overall, the observed upregulation of transplant-rejection-related pathways in this study suggests that Met may promote a microenvironment conducive to fat formation by modulating immune responses.

High concentrations of Met (>200 μM) did not exhibit cytotoxicity, as cell viability remained above 85%, suggesting that the diminished effect at higher concentrations may result from metabolic adaptation. Excessive Met can compete with other amino acids essential for lipid synthesis, impairing their uptake and inhibiting core differentiation pathways [51]. Furthermore, Met serves as a precursor for S-adenosylmethionine (SAM), a methyl donor. Excessive SAM can induce hypermethylation of adipogenic gene promoters, such as *PPARγ*, thereby suppressing transcription and lipid deposition [52,53,54]. These findings are consistent with the observed reduction in differentiation efficiency at higher Met concentrations.

In this study, RNA-seq data indicated that GO terms upregulated following Met supplementation primarily clustered around developmental growth, myotube organization, myofibril assembly, and molecular function, whereas downregulated terms concentrated on immunity. Adipocyte differentiation depends on the tissue microenvironment’s signaling, and muscle development directly affects intramuscular preadipocyte fate [4]. For example, rat muscle-derived IGF-1 activates preadipocyte proliferation [6], consistent with Met-induced proliferation of goat intramuscular preadipocytes; here, activin inhibits adipogenesis by suppressing *PPARγ* [7]. The upregulated “activin receptor antagonist activity” in our study suggests that Met may promote adipogenic differentiation by inhibiting activin receptors and relieving *PPARγ* suppression, aligning with our results. Research indicates that pro-inflammatory factors from activated natural killer cells/leukocytes suppress *PPARγ* [4,43,55,56]. Downregulated terms like “natural killer cell mediated cytotoxicity upregulation” in the Met group indicate that Met supplementation reduces excessive immune activation and pro-inflammatory factors, creating a favorable environment for intramuscular adipocyte differentiation. Sodium–iodide symporter (NIS, encoded by *SLC5A5*) inhibits fat deposition by promoting thyroid hormone synthesis (activates lipases) [57], consistent with our findings. In summary, Met indirectly supports intramuscular fat formation in goats through the following pathway: “microenvironmental support → maintenance of metabolic homeostasis → lifting of differentiation suppression → regulation of the immune microenvironment”.

In this study, RNA sequencing data revealed that *SLC5A5* expression decreased following the addition of Met. The SLC5 gene family comprises twelve members, including cotransporters responsible for the transport of sugars, anions, vitamins, and short-chain fatty acids [58]. *SLC5A5* is a sodium–iodide cotransporter located on the basal side; it is highly expressed in the thyroid gland, where it mediates accumulation of iodide, which is required for the biosynthesis of the thyroid hormones T3 and T4 [59]. In a study conducted on mice, it was discovered that the mRNA expression levels of *FABP4* exhibit a positive correlation with *SLC5A5* [60]. *FABP4* is a fatty acid transporter that coordinates lipid metabolism, and the expression of this gene promotes lipid deposition [11]. Research indicates that knocking out the *SLC5A5* gene in mice significantly increases their fat mass [57,61], suggesting that *SLC5A5* participates in adipogenic differentiation. Further studies revealed that knocking out the *SLC5A5* gene in mice promotes fat deposition by suppressing hormone-sensitive lipase (*HSL*) protein expression [62]. *HSL* is a pivotal enzyme that mediates triglyceride hydrolysis to release free fatty acids and glycerol from adipocytes in a hormonally controlled lipolysis process [63]. Consistent with this are previous findings that supplementing Met in the diet significantly increases the thickness of the backfat layer in beef cattle and promotes the deposition of body fat [64]. In this study, *SLC5A5* overexpression inhibited adipogenic differentiation in goat intramuscular adipocytes, reducing lipid droplet formation and the expression of *GLUT4*, *C/EBPβ*, and *PPARγ*, whereas Met supplementation reversed these effects. Conversely, *SLC5A5* silencing promoted adipogenic differentiation, and Met supplementation further enhanced this effect. These findings indicate that Met promotes adipogenic differentiation, at least in part, through downregulation of *SLC5A5*.

In summary, Met supplementation promotes proliferation and differentiation of goat intramuscular adipocytes in a dose-dependent manner, with optimal effects observed at 100 μM. This occurs via modulation of key signaling pathways, maintenance of a favorable immune microenvironment, and inhibition of *SLC5A5* expression.

## 5. Conclusions

This study demonstrates that intramuscular adipocyte proliferation and differentiation exhibit a quadratic relationship with Met concentration in basal complete medium, with an optimal concentration of 100 μM. Methionine promotes adipogenic differentiation in goats primarily by downregulating *SLC5A5*, which acts as a negative regulator of lipid deposition. These findings elucidate the molecular mechanism through which Met regulates IMF deposition, identify *SLC5A5* as a potential target for improving goat meat quality, and provide a theoretical basis for enhancing meat quality through nutritional modulation of intramuscular fat.

## Figures and Tables

**Figure 1 animals-15-03450-f001:**
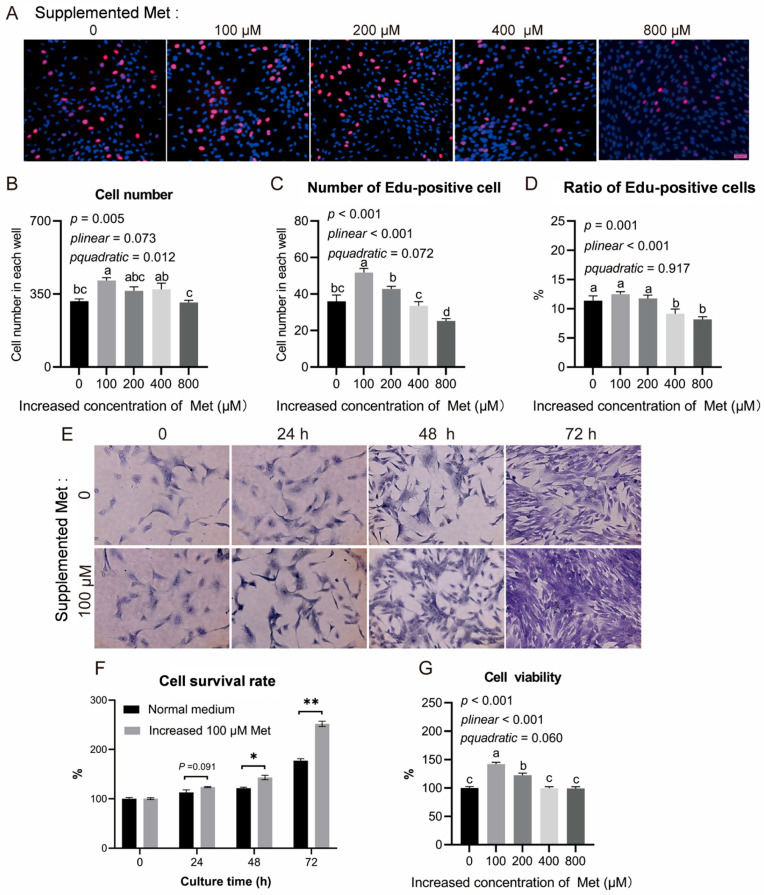
Cell proliferation and survival rate of intramuscular preadipocytes under different dosages of Met supplementation. (**A**–**D**) EdU staining and analysis of the EdU-positive cells under different dosages of Met supplementation. (**E**,**F**) Crystal violet staining analysis of cell viability under different culture periods. (**G**) CCK8 assay and analysis of the cell survival rate under different Met supplementation levels. Error bars, SEM (*n* = 4 wells). Scale bar, 50 μm. Labeled means with different letters are significantly different, *p* ≤ 0.05. * means *p* < 0.05, ** means *p* < 0.01.

**Figure 2 animals-15-03450-f002:**
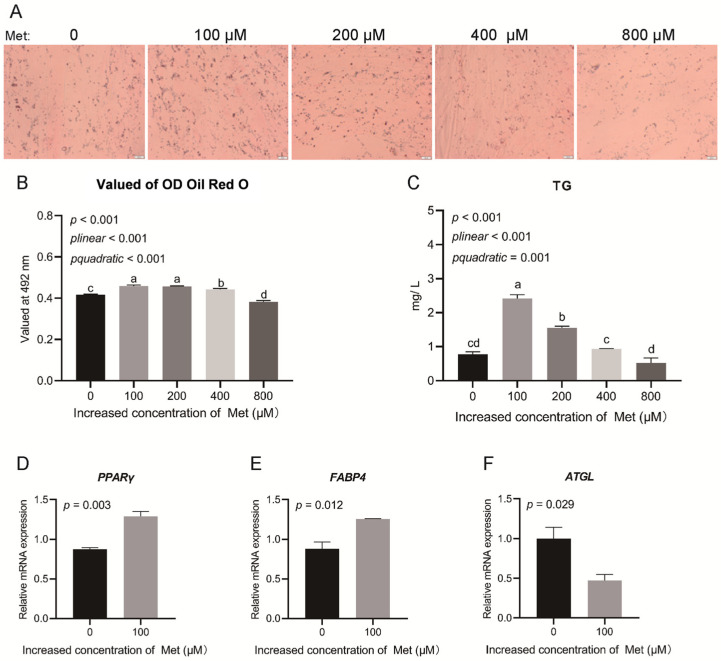
Cell differentiation and the expression of genes related to cell differentiation and lipid metabolism of intramuscular preadipocytes under different dosages of Met supplementation. (**A**,**B**) Oil Red O staining. (**C**) Triglyceride (TG) content detected by a commercial kit. Error bars, SEM (*n* = 4 wells). (**D**–**F**) Expression of genes related to cell differentiation and lipid metabolism during Met supplementation in intramuscular preadipocytes. Scale bar, 20 μm. Error bars, SEM (*n* = 3 wells). Labeled means with different letters are significantly different, *p* ≤ 0.05.

**Figure 3 animals-15-03450-f003:**
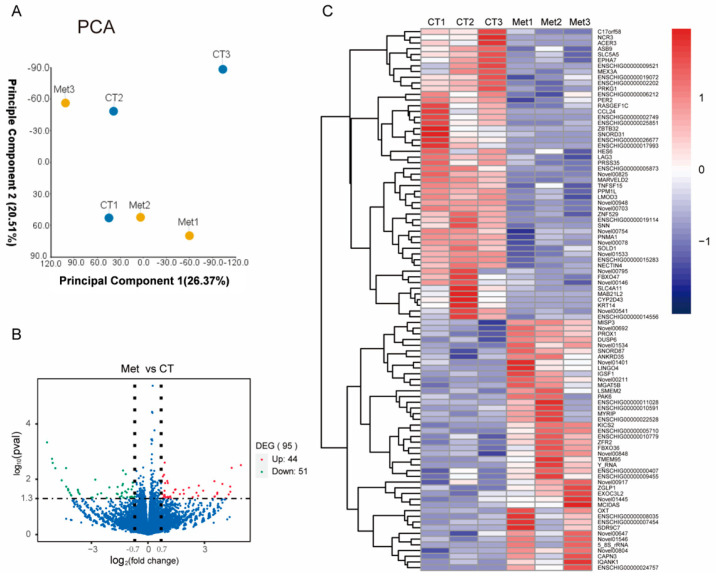
PCA analysis (**A**), volcano figure (**B**), and heatmap (**C**) of RNA sequencing data. CT, the basal complete medium. Met, the basal complete medium supplemented with 100μM Met. DEG, different expressed gene (*n* = 3 wells). Genes in the heatmap were selected from DEGs for which *p* ≤ 0.05, |log2(−foldchange)| ≥ 0.7. Each treatment had three replicates.

**Figure 4 animals-15-03450-f004:**
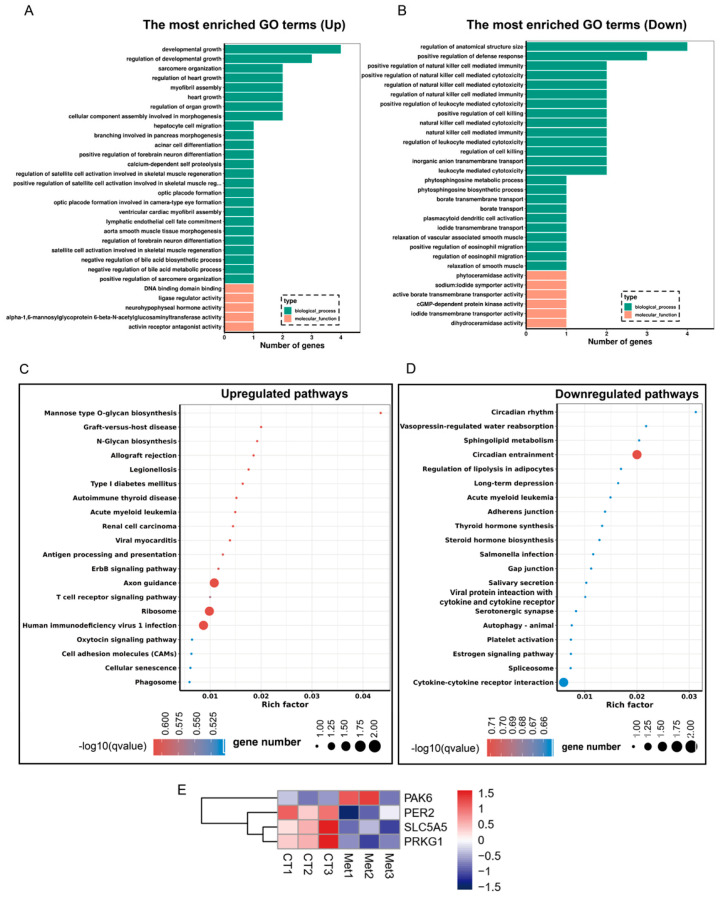
GO and KEGG enrichment for DEGs. (**A**) The top 30 significant enrichment upregulated GO terms are listed by q-value. (**B**) The top 30 significant enrichment downregulated GO terms are listed by q-value. (**C**) The top 20 significant enrichment upregulated pathways are listed by q-value. (**D**) The top 20 significant enrichment downregulated pathways are listed by q-value. (**E**) Key potential transcription factors. *n* = 3 wells.

**Figure 5 animals-15-03450-f005:**
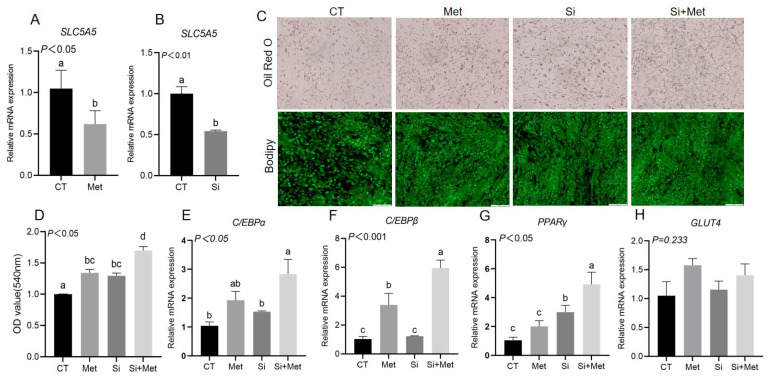
The effect of Met supplementation following *SLC5A5* silencing on the differentiation of goat intramuscular adipocytes. (**A**) The effect of supplemental Met on the transcription factor *SLC5A5*. (**B**) *SLC5A5* knockdown efficiency detection. (**C**) Oil Red O staining and Bodipy staining. (**D**) The OD value of Oil Red O. (**E**–**H**) Supplementation with Met and interference of *SLC5A5* alter mRNA expression of adipose deposition and adipocyte differentiation marker genes. Student’s *t*-test (*n* = 3 wells). Different letters on the bar graph indicate significant differences, *p* < 0.05.

**Figure 6 animals-15-03450-f006:**
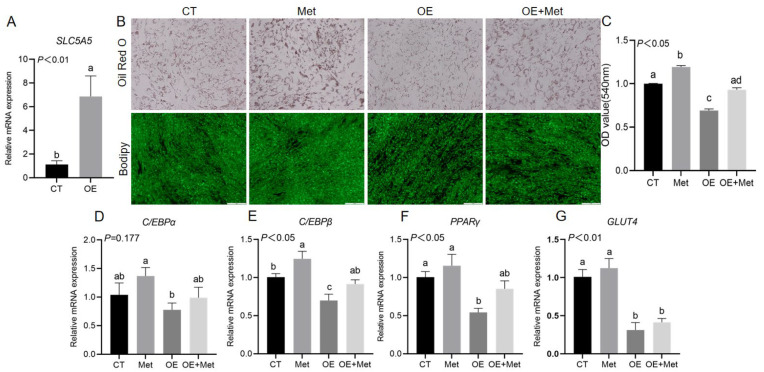
The effect of Met supplementation on the differentiation of goat intramuscular adipocytes following *SLC5A5* overexpression. (**A**) Detection of *SLC5A5* overexpression efficiency. (**B**) Oil Red O and Bodipy staining. (**C**) The OD value of Oil Red O. (**D**–**G**) Supplementation with Met and overexpression of *SLC5A5* alter mRNA expression of adipose deposition and adipocyte differentiation marker genes. Student’s *t*-test (*n* = 3 wells). Different letters on the bar graph indicate significant differences, *p* < 0.05.

**Table 1 animals-15-03450-t001:** Sequences of primers.

Primer Name	Accession Number	Primer Sequence	Tm/°C	Product Length/bp
*GADPH*	NM_001285607.1	F: GGTCGGAGTGAACGGATTTGG R: CATTGATGACGAGCTTCCCG	60	197
*PPARγ*	NM_001285658.1	F: AAGCGTCAGGGTTCCACTATG R: GAACCTGATGGCGTTATGAGAC	60	197
*FABP4*	NM_001285623.1	F: GAAAGAAGTGGGTGTGGGCT R: TGGTGGTAGTGACACCGTTC	60	317
*ATGL*	NM_001285739.1	F: GGAGCTTATCCAGGCCAATG R: TGCGGGCAGATGTCACTCT	60	180
*C/EBPα*	XM_018062278.1	F: CCGTGGACAAGAACAGCAAC R: AGGCGGTCATTGTCACTGGT	58	142
*C/EBPβ*	XM_018062278.1	F: AAGAAGACGGTGGACAAGC R: AACAAGTTCCGCAGGGTG	65	204
*GLUT4*	NM_001314227.1	F: TGCTCATTCTTGGACGGTTCT R: CATGGATTCCAAGCCTAGCAC	59	176

## Data Availability

Data are contained within the article.

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
