# Peer review of "Methionine Supplementation Benefits Lipogenesis in Goat Intramuscular Adipocytes, Likely by Inhibiting the Expression of SLC5A5"

_animals, 2025, doi:10.3390/ani15233450_

Round 1

Reviewer 1 Report

Comments and Suggestions for Authors

This study investigates the effects of methionine (Met) on the proliferation and differentiation of goat intramuscular adipocytes and explores the underlying molecular mechanisms. The novelty lies in revealing the mediating role of SLC5A5 in Met-regulated adipogenesis. The experimental design is systematic, but the language expression, data presentation, and interpretation of some results require significant improvement.

1.The manuscript contains numerous grammatical errors, spelling mistakes, and unnatural expressions, which severely affect readability. For example, “Intransucular processor adipocytes” should be corrected to “Intramuscular preadipocytes,” and “underling the reason” should be “underlying reason.” It is recommended that the text be thoroughly polished by a native English speaker or professional editing service.

2.The terminology for "preadipocytes" is inconsistent, with multiple terms such as “precursor adipocyte,” “processor adipocyte,” and “preadipocyte” used interchangeably. These should be unified as “preadipocyte.”

3.The abstract and conclusions are highly repetitive. It is recommended that the abstract briefly summarize the main findings, while the conclusions provide a more in-depth summary and future perspectives.

4.The introduction lacks sufficient background on the known functions of the SLC5A5 gene in lipid metabolism. Its roles in thyroid hormone metabolism and lipid metabolism should be supplemented to highlight the novelty of this study.

5.The rationale for the study could be stronger. Although the role of Met in fat accumulation is mentioned, there is a lack of discussion on the specific response of goat intramuscular adipocytes. The connection between this research and the practical need to improve goat meat quality should be emphasized.

6.It is not clearly stated whether the basal culture medium contains Met, which may affect the interpretation of the "Met supplementation" effects.

7.The description of the RNA-seq methodology is incomplete. Key details such as library construction, sequencing platform, and data quality control are missing and should be supplemented.

8.The siRNA sequences used for SLC5A5 knockdown are not provided. Either the sequences or relevant references should be included.

9.The clarity of Figures 4A–E is low, making it difficult to read the axis labels. Please provide higher-resolution images.

10.In Figures 4A and 4B, many upregulated GO terms are not directly related to adipogenesis. The potential significance of these terms or whether they represent background noise should be discussed. Similarly, in Figure 4C, several upregulated pathways (e.g., graft-versus-host disease, autoimmune diseases) have unclear relevance to adipocyte biology and require reasonable explanation or cautious interpretation.

11.In Figures 5 and 6, Met still exerts effects under SLC5A5 knockdown or overexpression, suggesting the existence of SLC5A5-independent pathways. The authors should discuss this possibility rather than focusing solely on the SLC5A5-dependent mechanism.

12.The discussion overly focuses on SLC5A5, while the RNA-seq data reveal other potentially important genes and pathways (e.g., PER2, PRKG1). These factors should be incorporated into the discussion to construct a more comprehensive mechanistic network.

13.The discussion on the role of SLC5A5 in lipid metabolism primarily cites studies in mice. There is a lack of supporting literature in ruminants or intramuscular adipocytes.

14.Why is 100 μM the optimal concentration? The potential reasons for the reduced effects at concentrations above 200 μM (e.g., toxicity or metabolic alterations) are not explored.

15.It is recommended that the RNA-seq DEG list and complete GO/KEGG enrichment results be provided as supplementary materials to enhance data transparency.

Reviewer 2 Report

Comments and Suggestions for Authors

This study addresses the topical issue of methionine's effect on adipocyte lipogenesis, which has implications for animal husbandry. The experimental design is generally sound, and the findings are potentially significant. However, the manuscript requires significant revision to improve clarity, eliminate numerous linguistic and stylistic errors, and strengthen the scientific validity and interpretation of some of the results.

The research question is relevant. Studying the effect of methionine on intramuscular adipocytes in goats and the role of the SLC5A5 transporter in this process is somewhat novel, especially in the context of this species. The clearly formulated hypothesis that methionine promotes adipogenesis through the suppression of SLC5A5 represents a concrete contribution to this field. The obtained results expand our understanding of the mechanisms regulating meat quality.

  1. Methionine at an optimal concentration (100 μM) stimulates adipocyte proliferation and differentiation, and this effect appears to be mediated by inhibition of SLC5A5. However, the interpretation of some data requires further analysis. For example, the statement that knockdown of SLC5A5 promotes adipogenesis , and methionine abolishes this stimulatory effect (lines 39-41 of the abstract), seems contradictory to the main conclusion and requires clear clarification. Are all the conclusions fully supported by the results? Generally yes, but some of the wording may be misleading.
  2. The study was adequately designed: a range of Met concentrations were used, standard analysis methods (staining, qPCR, RNA- seq ) were applied, and functional tests ( gene overexpression /knockout) were performed. However, the description of the methods is insufficiently detailed to ensure reproducibility.
  3. It is unclear how exactly cell proliferation was assessed using crystal violet (section 2.6)—was it quantified? The siRNA sequence for SLC5A5 is not provided.
Comments on the Quality of English Language

The text requires full professional linguistic editing

Reviewer 3 Report

Comments and Suggestions for Authors

This manuscript investigates the role of methionine (Met) in the proliferation and differentiation of goat intramuscular adipocytes, with a focus on identifying underlying molecular mechanisms through RNA-seq and functional validation of SLC5A5. The study is of potential interest to the field of animal science and meat quality improvement. However, several methodological and presentation issues need to be addressed before the paper can be considered for publication. Below are specific observations and suggestions for improvement.

Suggestions

Line 91 and throughout the manuscript.The term Longissimus dorsi is outdated and not anatomically correct in veterinary terminology. Please use the correct term from the Nomina Anatomica Veterinaria (NAV, 6th edition, 2017) — either Longissimus thoracicus or Longissimus lumborum — depending on the actual sampling site. Reference: Nomina Anatomica Veterinaria (WAVA). Additionally, please clarify how many samples were used for tissue collection and biological replicates.

Lines 165–170 (qRT-PCR assay section). The qRT-PCR methodology lacks essential details. Please describe: The positive control used and negative control.

Line 181 (Table 1: Primer sequences). Please revise Table 1 to include the following information for each primer pair: Melting temperature (Tm), Target gene name and Source or citation for primer design (if adapted from a previous publication, please cite it).

Figures 1, 2, 5, and 6 - Each figure should clearly indicate the sample size (n) used in each experimental group and the statistical test applied (e.g., one-way ANOVA, Tukey’s post hoc test, Student’s t-test, etc.). 

Figures 3C and 4 - These figures are difficult to interpret due to poor image quality or low contrast. Please provide higher-resolution or more clearly labeled figures to facilitate data interpretation. Ensure that all scale bars and labeling are legible and consider enhancing contrast or adjusting color balance for better visualization of key features.

The manuscript presents promising results that contribute to understanding methionine’s regulatory role in intramuscular fat deposition. 

Round 2

Reviewer 1 Report

Comments and Suggestions for Authors

The authors have adequately addressed all my major concerns. The manuscript has shown significant improvement in language, methodological details, interpretation of results, and depth of discussion.

Reviewer 2 Report

Comments and Suggestions for Authors

The article has been significantly revised and has become much stronger, especially due to the clear presentation of the functional data on SLC5A5, which convincingly support the main hypothesis. The critical contradiction noted by the reviewer has been corrected.

Section 3.6 clearly demonstrates that knockdown of SLC5A5 promotes adipogenesis, and that addition of methionine further enhances this effect, which is logical and supports the hypothesis.

However, a new problem with interpretation has emerged. In the same section (3.6), the authors call this effect "synergistic." This is terminologically incorrect. If methionine acts by suppressing SLC5A5, then adding methionine to a gene already "silenced" should not produce a strong additional effect. The observed significant effect may indicate that methionine affects not only SLC5A5 but also other targets. The authors should reformulate this as an "additive effect" or describe the result more carefully, without resorting to the term "synergism," or discuss the possibility of additional mechanisms.
